# Recent Perspectives in the Management of Fungal Keratitis

**DOI:** 10.3390/jof7110907

**Published:** 2021-10-26

**Authors:** Nimmy Raj, Murugesan Vanathi, Nishat Hussain Ahmed, Noopur Gupta, Neiwete Lomi, Radhika Tandon

**Affiliations:** 1Cornea, Lens & Refractive Surgery Services—Dr R P Centre for Ophthalmic Sciences, All India Institute for Medical Sciences, New Delhi 110029, India; nimmyraj19@gmail.com (N.R.); noopurparakh@gmail.com (N.G.); neiwete@gmail.com (N.L.); radhika_tan@yahoo.com (R.T.); 2Ocular Microbiology Services—Dr R P Centre for Ophthalmic Sciences, All India Institute for Medical Sciences, New Delhi 110029, India; drnishathussain@gmail.com

**Keywords:** mycotic, fungal keratitis, refractory keratitis, metagenomic deep sequencing, polyenes, azoles, echinocandins, repeat culture, therapeutic keratoplasty, tear genomics, PCR, antifungal susceptibility testing

## Abstract

Mycotic keratitis is common in warm, humid regions with a varying profile of pathogenic fungi according to geographical origin, socioeconomic status, and climatic condition. Clinical diagnosis can be challenging in difficult cases and those refractory to treatment. Fungal hyphae on microscopic examination and culture isolation have been the gold standard in the laboratory diagnosis of fungal keratitis. A culture isolate of the aetiological fungus is essential to perform antifungal susceptibility testing. As the culture isolation of fungi is time-consuming, causing delays in the initiation of treatment, newer investigative modalities such as in vivo confocal microscopy and molecular diagnostic methods have recently gained popularity. Molecular diagnostic techniques now help to obtain a rapid diagnosis of fungal keratitis. Genomic approaches are based on detecting amplicons of ribosomal RNA genes, with internal transcribed spacers being increasingly adopted. Metagenomic deep sequencing allows for rapid and accurate diagnosis without the need to wait for the fungus to grow. This is also helpful in identifying new emerging strains of fungi causing mycotic keratitis. A custom-tear proteomic approach will probably play an important diagnostic role in future in the management of mycotic keratitis. Positive repeat cultures are being suggested as an important gauge indicative of a poor prognosis. Positive repeat fungal cultures help to modify a treatment regimen by increasing its frequency, providing the addition of another topical and oral antifungal agent along with close follow-up for perforation and identifying need for early therapeutic keratoplasty. The role of collagen crosslinking in the treatment of fungal keratitis is not convincingly established. Rapid detection by multiplex PCR and antifungal susceptibility testing of the pathogenic fungi, adopted into a routine management protocol of fungal keratitis, will help to improve treatment outcome. Early therapy is essential in minimizing damage to the corneal tissue, thereby providing a better outcome. The role of conventional therapy with polyenes, systemic and targeted therapy of antifungal agents, newer azoles and echinocandins in fungal keratitis has been widely studied in recent times. Combination therapy can be more efficacious in comparison to monotherapy. Given the diversity of fungal aetiology, the emergence of new corneal pathogenic fungi with varying drug susceptibilities, increasing the drug resistance to antifungal agents in some genera and species, it is perhaps time to adopt recent molecular methods for precise identification and incorporate antifungal susceptibility testing as a routine.

## 1. Introduction

Corneal blindness is responsible for about 1.5 to 2 million new cases of monocular blindness every year, with ocular trauma and infectious keratitis being accountable for the majority of cases [1,2]. Fungal keratitis constitutes about 50% of all culture-positive cases of infective keratitis in developing countries [3,4]. It is common in areas with warm and humid climates and among populations mainly engaged in agricultural activities. The common cornea pathogenic fungi include *Aspergillus* spp., *Fusarium* spp. and *Candida* spp., with regional variations depending on geographical, occupational and host factors [5]. Corneal infections due to filamentous fungi are found to predominantly cause infection in tropical and subtropical regions, and yeasts more frequently in temperate climates [5,6]. The most common fungi isolated from our centre included *Aspergillus* spp. (31.1%), followed by *Fusarium* spp. (24.5%), *Alternaria* (10.5%)*, Curvularia* (10.2%), *Helminthosporium* (5.7%), *Bipolaris* (5.4%), *Penicillium* (4.5%), *Candida* (4.4%), *Acremonium* (1.2%), *Rhizopus* (1.0%), *Paecilomyces* (0.8%), *Rhodotorula* (0.5%) and *Mucor* (0.2%) [7].

Characteristic clinical features of mycotic keratitis include dry-looking corneal ulcers with satellite lesions, associated with hypopyon and endothelial plaque. The demonstration of fungal elements in microscopic examination and culture has been the gold standard for diagnosing fungal keratitis. As the culture isolation of fungi is time consuming, causing delays in the initiation of treatment, newer investigative modalities such as in vivo confocal microscopy [8,9] and molecular diagnostic methods have gained popularity recently as enabling rapid and real-time diagnosis [10,11].

Understanding the pathogenesis of the disease, as well as early diagnosis and prevention strategies, plays a significant role in the successful management of fungal keratitis [12]. Ideal therapeutic effects may not be achieved through the currently available medical therapeutic options. With very few commercially available topical antimycotic agents in several parts of the world, there is a need to reconstitute the antifungal therapeutic agents for effective topical therapy in refractory cases. There is also the rising concern of resistance to topical antifungal therapeutic agents which has led to the increased adoption of antifungal susceptibility testing (AFST) in the management of fungal keratitis in recent times. Poor ocular bioavailability, limited retention time and low ocular tissue penetration are areas of major concern in the topical use of anti-fungal agents. Delays in presentation, the injudicious use of steroid drops, the deep penetration ability of the organisms, challenges in diagnosis with a poor yield of organisms from scraping specimens, varying response to antifungal treatment and emerging resistance to commonly used agents and the unavailability of appropriate anti-fungal agents are responsible for poor treatment outcomes in fungal keratitis. The successful management of fungal keratitis can be challenging, with a high requirement for surgical interventions. The need for therapeutic penetrating keratoplasty (TPK) in fungal keratitis has been reported to be as high as 50% [13,14]. In this review, we discuss recent concepts in the management of mycotic corneal infections with relevance to current perspectives on the role of newer investigative imaging and molecular diagnostic modalities, repeat cultures, antifungal susceptibility testing (AFST), antimycotic therapeutic agents and the outcomes of corneal collagen crosslinking (CXL) and TPK in fungal corneal ulcers.

## 2. Diagnosis

### 2.1. Clinical Diagnosis

The clinical diagnosis in a case of fungal keratitis involves detailed history-taking, considering the mode of injury, occupation of the patient, treatment taken, use of steroids if any and any similar illness in the past. The clinical signs are usually much more than symptoms in a case of fungal keratitis. On slit lamp biomicroscopic examination, a dry-looking ulcer with stellate feathery or irregular margin with associated satellite lesions (Figure 1) is the usual presentation. Depending on the aetiological agent, there can be endothelial plaques and pigmentation and a fixed hypopyon. The reported diagnostic accuracy of fungal keratitis by clinical examination only has been reported to be less than 70% [15], hence accurate diagnosis using microscopic examination, culture, in vivo confocal microscopy and molecular diagnostic methods becomes imperative for early identification.

### 2.2. Laboratory Diagnosis

Laboratory diagnosis with culture isolation in fungal keratitis is of extreme importance to provide appropriate and successful treatment. It also helps in performing antifungal susceptibility testing to establish the sensitivity to treatment with conventional and newer antifungal agents. Specimens collected for microbiological evaluation include corneal scrapings in most cases, or corneal biopsy or suture biopsy in cases with deeper infiltrates or refractory cases where routine specimens fail to provide a yield. Conventional methods of diagnosis of fungal keratitis included staining of the smear and culture of the corneal scraping. Recently, molecular diagnostic methods like polymerase chain reaction are gaining popularity as they allow accurate and rapid diagnosis. Real time in vivo confocal microscopy corneal imaging can detect fungal hyphae to establish an early diagnosis in fungal keratitis.

#### 2.2.1. Microscopic Examination

The detection of fungal elements in collected corneal scraping specimens on microscopic examination has been the conventional mode of diagnosing keratitis. Stains used to detect fungal organisms include Gram, 10% potassium hydroxide (KOH) wet mount, lactophenol cotton blue, Giemsa, calcofluor white, Grocott–Gomori’s methanamine silver (GMS), acridine orange and periodic acid Schiff (PAS) stains [16,17]. Detection of fungal hyphae in KOH mounts (Figure 2) is the most commonly used procedure to reach a provisional diagnosis in many parts of the world due to its cost effectiveness, simplicity of procedure, easy availability and ability to provide quick results which can help in prompt initiation of empirical therapy. The sensitivity and specificity of KOH mounts have been found to range from 60% [18,19] to 99.3% [20], and from 70% [18,21] to 99.1% [20], respectively. The sensitivity of various staining techniques will vary depending on the type of stain used (KOH, Gram, acridine-orange, calcofluor), the experience of the microbiologist and the quantity and type of sample. In situations where corneal smear staining and culture does not yield any organism, the progression of the disease despite maximal treatment is seen or when corneal involvement is much deeper and scraping does not seem to be a feasible option, a corneal biopsy can be considered. The biopsy specimen is sent for both culture and histopathology. The use of special stains like PAS or GMS can better delineate the fungal hyphae and yeast. Some studies [22] have shown corneal biopsy specimens to have higher sensitivity than scraping samples which can be attributed to factors such as deep stromal involvement by a few fungi or can be due to the reduced amount of corneal sample obtained on scraping.

#### 2.2.2. Culture

Culture growth of the causative fungus, though time consuming, is essential for species identification and successful treatment. Furthermore, this helps to perform antifungal susceptibility testing and establish the sensitivity to treatment with conventional and newer antifungal agents. The results of culture are highly specific and it is thus considered the gold standard in the diagnosis of fungal keratitis. The commonly used culture media include Sabouraud’s dextrose agar (SDA) (incubation at 22–25 °C), (Figure 3) potato dextrose agar, blood agar (incubation at 37 °C), chocolate agar and thioglycolate agar. Cycloheximide inhibits the growth of fungus and should not be present in the media used for culture. A C-steak pattern in solid media is used and the culture is checked for growth daily. A positive result is one in which the organism grown on two or more solid media is the same, when there is semiconfluent growth at the site of inoculation and when the organism grown is similar to the one identified with staining techniques [20,23]. The culture usually takes up to more than one week to show growth and this can possibly delay the diagnosis. The low sensitivity rates, the need for an experienced microbiologist to interpret the results and the inability to differentiate between species with morphologically similar growth are several other limitations.

The inability to detect an organism with the conventional methods (clinical diagnosis, smears and culture) can lead to inadequate treatment resulting in non-healing keratitis and subsequent vision loss. A repeat culture, after the initial treatment initiation, is now being endorsed as a new diagnostic and prognostic tool. A secondary analysis of the mycotic ulcer treatment trial (MUTT)-1 (milder and early ulcers) [24] and MUTT-2 (severe fungal ulcers) [25] data showed positive reports on repeat cultures done at six days of initiation of treatment to be associated with poorer 3-month visual acuity, larger scar size and an increase in the rate of perforation and/or the need for therapeutic penetrating keratoplasty (TPK). Hence a repeat culture at day six can act as an important prognostic tool indicating the need to closely follow up, increase the dosage or add newer agents like systemic drugs or a different group of drugs to an ongoing treatment regimen and consider early surgical intervention in the form of TPK or lamellar procedures [26,27] like therapeutic deep anterior lamellar keratoplasty (DALK) in those patients found positive. Repeat cultures can help distinguish the effectiveness of antifungal agents, both conventional and newer ones, as it can reflect on its fungicidal activity. Hence, sixth-day cultures are now being recommended as an important prognostic indicator [28] and more trials done in future in this direction will perhaps help to further establish this.

#### 2.2.3. In Vivo Confocal Microscopy

In vivo confocal microscopy (IVCM) is a novel non-invasive technique to analyse corneal cellular structure layer by layer with the help of real-time microscopic images. With this technology each layer of the cornea can be delineated as in in vitro histochemical techniques. In fungal keratitis, due to its non-specific clinical features, poor yields on scraping specimens giving no conclusive results, the variable sensitivity of culture results [4,20] and the prolonged time taken for culture to show growth, the diagnosis is often delayed, affecting the institution of appropriate treatment and consequent poor treatment outcomes. IVCM helps in overcoming most of these limitations and also has the added advantage of being non-invasive in nature. In addition, it can help ascertain the depth of involvement [9] and also help in assessing the efficacy of antifungal agents.

Kanavi et al. [29] in their study on the use of IVCM in infectious keratitis observed a 94% sensitivity and 78% specificity for fungal keratitis using a tandem scanning in vivo confocal microscope (TS-IVCM). Similarly, Vaddavalli et al. [30] in their study of keratitis found a sensitivity of 89.2% and a specificity of 92.7% in fungal keratitis cases using a slit scanning confocal microscope in fungal and acanthamoeba keratitis. Chidambaram et al. [8] utilised a laser scanning confocal microscope in infective keratitis cases and obtained a pooled sensitivity of 85.7% and a pooled specificity of 81.4% in fungal filament detection.

Fungal organisms are diagnosed in IVCM if parallel lines of high-contrast elements (Figure 4) resembling *Fusarium* hyphae (branching at 90 degree), *Aspergillus* hyphae (branching at acute angles) or *Candida* pseudo filaments in the anterior stroma are visualised [31]. Chidambaram et al. [32] from their study, which included 183 cases of fungal keratitis, have described inflammatory cells in a honeycomb pattern in the anterior stroma (found in 49% of cases) with absence of stromal bullae as typical IVCM features of fungal keratitis. Besides the limitations of this technique (being a contact procedure and the level of cooperation from the patient required to perform the test in the highly symptomatic stage of the disease), it is an important tool in diagnosing certain kinds of microbial keratitis. Its high cost, limited accessibility and inability to diagnose organisms at the species level restrict its use as a primary method of diagnosis.

#### 2.2.4. Molecular Diagnostic Methods

Molecular diagnostic techniques now help to obtain a rapid diagnosis of fungal keratitis. The techniques are mainly polymerase chain reaction (PCR)-based and help overcome the lower sensitivity of conventional laboratory techniques while maintaining the specificity. In this technique, the target gene of interest, called the nucleic acid template, is amplified in a thermo-cycling reaction and by virtue of it, billions of copies are produced from a single template. This enables the clinician to reach a diagnosis even if the collected sample size is small or has no viable fungal elements. Molecular characterisation with techniques like post amplification sequence analysis can identify all fungi up to species level and detect rare emerging pathogenic fungi which are difficult to diagnose with conventional methods alone. Different molecular methods used for the diagnosis and/or identification of causative agents in fungal keratitis include conventional PCR, nested PCR, multiplex PCR [11,33], real-time PCR [34,35], conventional PCR followed by enzymatic digestion/dot hybridization/sequencing/single strand conformation polymorphism (SSCP) [36,37,38], high resolution melting analysis and next-generation sequencing combined with computational analysis [39]. Manikandan et al. [11] compared the results of culture-proven fungal keratitis from corneal scrapings with multiplex PCR and found agreement in 94.1% of *Fusarium*, 100% of *Aspergillus flavus* and 63.6% of cases of *Aspergillus fumigatus*. The minimum amount of fungal DNA to get a positive result in PCR examination in this study [11] was found to be 10 fg/μL, 1 pg/μL and 300 pg/μL of DNA for *Fusarium*, *A. flavus* and *A. fumigatus*, respectively. Oechsler et al. [40] emphasised the importance of exact species identification with molecular methods like PCR for optimum treatment and better treatment outcomes. Application of recent diagnostic methods will help to enhance the precision of diagnosis in fungal corneal infections.

Recently, metagenomic deep sequencing (MDS), which involves both DNA and RNA sequencing, has been suggested to have potential for improved diagnostic sensitivity and accuracy [41]. This allows for rapid diagnosis and helps obtain an accurate diagnosis without the need to wait for the fungus to grow. Lalitha et al. [42] recently published their experience with MDS in 46 corneal ulcer cases. With the help of latent classic analysis (LCA), they evaluated the sensitivity and specificity of conventional diagnostic tests, DNA sequencing and RNA sequencing. The sensitivity was 70% for KOH/Gram stain, 52% for culture and 74% for MDS. On LCA, RNA sequencing was found to be 100% sensitive and specific for bacterial keratitis and 100% sensitive and 97% specific for fungal cases [42]. MDS, though not FDA approved as yet, helps distinguish a causative pathogen from a colonized or contaminated pathogen.

A tear proteomic approach has also been hailed as providing comprehensive data on ocular surface defence and damage caused in fungal keratitis patients. A custom tear proteomic approach will probably play an important diagnostic role in the future in the management of mycotic keratitis. Recently recommended omics approaches [43], such as those using genomic, metagenomic and tear proteomic data sources, provide greater hope for better diagnosis and follow-up of fungal keratitis. Genomic approaches are based mainly on detecting amplicons of ribosomal RNA genes, with internal transcribed spacers being increasingly adopted in clinical practices. The recent sophisticated metagenomic approach is based on 16S rRNA genes to help monitor the dynamic change in conjunctival microbial flora associated with fungal keratitis [44,45]. Diagnostics based on 18S rRNA target enrichment sequencing in clinical samples have been suggested to have good potential to diagnose fungal corneal infections [46,47].

Matrix-assisted laser desorption ionization time-of-flight mass spectrometry (MALDI-TOF MS) is a steadfast technique which helps identify pathological organisms in minutes [48]. Earlier used only in the diagnosis of bacterial organisms, it has now emerged as a tool to identify isolates of fungi, particularly yeasts, and also a few genera of filamentous fungi, including *Aspergillus, Penicillium*, *Fusarium*, and *Mucorales* [49]. A recent study from a tertiary centre [50] in India found that MALDI-TOF MS used to specifically identify the aetiological spectrum of infective keratitis was able to accurately identify in 51% of cases (100% of culture positive), except 2% of polymicrobial growths. These newer modalities can help identify the exact species involved in the infection which in turn helps to commence appropriate treatment and pave the way for appropriate antifungal susceptibility testing [51]. Future strategies to reduce the morbidity associated with infectious keratitis are likely to be multidimensional, with adjuvant therapies aimed at modifying the immune response of the host to infections. These seem to hold the greatest potential to improve clinical outcomes.

#### 2.2.5. Antifungal Susceptibility Testing

In recent times, with increasing antifungal resistance and the introduction of new antifungal agents, AFST and minimum inhibitory concentration (MIC) determination have a very significant role in terms of the successful management of fungal keratitis. The aim of performing AFST is to provide practical data for the treating clinician on the susceptibility, dose-dependent susceptibility or resistance phenotype for an organism–antifungal agent combination. The antifungal agent of choice for certain fungi can be empirically assumed by the proper identification of the pathogen and may not always require susceptibility testing. These tests are most useful in cases of invasive fungal infections, when acquired drug resistance is presumed, and in refractory cases not responding to therapy. The two universally recognized standard method bodies, the Clinical and Laboratory Standards Institute (CLSI) [52,53] and the European Committee for Antimicrobial Susceptibility Testing (EUCAST) [54,55] have put forward phenotypic assays to perform in vitro AFST based on the broth micro-dilution method for both filamentous fungi and yeasts. The antifungal activity is expressed as MIC values of an anti-fungal drug, which refers to the minimal drug concentration that completely inhibits fungal growth. CLSI have laid down guidelines on MIC breakpoint values for candida species [56] and EUCAST on breakpoints for amphotericin B against *A. fumigatus* and *A. niger*, isavuconazole breakpoints against *A. fumigatus*, *A. nidulans* and *A. terreus*; itraconazole breakpoints against *A. flavus*, *A. fumigatus*, *A. nidulans* and *A. terreus*; posaconazole breakpoints against *A. fumigatus* and *A. terreus* and voriconazole breakpoints against *A. fumigatus* [57,58]. However, the data on the MIC breakpoints of other pathological fungi and antifungal agents are largely lacking. The different methods for susceptibility testing are broth micro-dilution for yeasts, disk diffusion methods, commercial alternatives like gradient diffusion strips, Sensititre Yeast One Assay and Vitek 2 Yeast Panels [51]. Gradient diffusion strips (E test) (Figure 5) are an effortless substitute for the broth micro-dilution and disk diffusion methods due to its ease of use [51], ability to give an MIC value which has a relatively good essential agreement (EA) with the standard methods [59,60] and the advantage of providing a much wider and diverse range of MIC values to discriminate between amphotericin B-susceptible and resistant isolates for both *Candida* and *Cryptococcus* spp. [61,62] which is not given by the standard methods.

AFST, performed by clinical microbiology laboratories as a tool to aid in the selection of the appropriate antifungal agent, has been reported to have a linear correlation between susceptibility and treatment outcome [63,64,65]. Sun et al. [66] prospectively assessed the effect of MIC on clinical outcomes during the course of treatment with a single agent administered by a standardized protocol with the data from the MUTT-1 trial. They found that in natamycin-treated cases, a twofold increase in MIC was significantly associated with a larger 3-month infiltrate/scar size and increased odds of perforation, but was not associated with 3-month visual acuity. No correlation could be found with the voriconazole group. This study also noted 92% of Fusarium isolated was sensitive to natamycin with an MIC value <32 μg/mL. Lalitha et al. [64], with the same data from the MUTT-1 trial, demonstrated that natamycin had higher MIC values against all isolates except *Fusarium* spp. whereas voriconazole had the lowest MIC against *Aspergillus* species. Their study also showed a high MIC to be associated with higher odds for perforation. Saha et al. [67] in their study from eastern India assessed the antifungal susceptibility of fungal isolates to commonly used antifungal agents using the disk diffusion method and found *Aspergillus* and *Fusarium* to be more sensitive to voriconazole than natamycin, and amphotericin B to be effective against *Candida*. AFST, though not routinely recommended, can have an important role in the management of recalcitrant mycotic keratitis and in clinical scenarios when rare organisms/emerging new pathogens are identified on culture. Further studies in this direction can have a profound impact on the comprehensive management of fungal keratitis.

## 3. Treatment

Antifungal agents for ophthalmological use are given as topical formulations/systemic agents orally or local injection like in targeted therapy with intra cameral and intracorneal injections. They predominantly belong to the following classes of drugs: polyenes (amphotericin B, natamycin, nystatin), azoles/imidazoles (ketoconazole, miconazole, econazole), triazoles (itraconazole, voriconazole, posaconazole, fluconazole, ravuconazole) and echinocandins (caspofungin, micafungin, anidulafungin).

### 3.1. Topical Agents

Polyenes and azoles are a mainstay of the topical treatment of corneal mycotic infections. Natamycin is the only anti-fungal formulation that has been US-FDA approved for treating ocular fungal infections but off-label use of other agents is common in clinical settings. The treatment of a case of fungal keratitis is often prolonged with a majority of the cases requiring weeks to months for complete resolution of symptoms. The most commonly used topical anti-fungal agents are natamycin 5% drops for filamentous fungi and amphotericin 0.15% for yeast-like fungus [27]. Azoles, both imidazoles and triazoles, are used as either adjunctive or alternative agents in non-responding and recalcitrant cases. New generation azoles like voriconazole are now being increasingly used in the management of fungal keratitis due to its broad-spectrum and better ocular penetration profile.

### 3.2. Systemic Therapy

The intermittent dosing of topical medications may result in intervals of suboptimal drug levels and oral medications can help provide more steady-state drug levels to overcome this limitation of exclusive topical medications [58]. The systemic antifungal agents in common practice include ketoconazole, itraconazole, fluconazole and voriconazole. However, the role of oral antifungal therapy in the management of kerato-mycosis is still inconclusive. The MUTT-2 trial [14], the only randomized control trial done to evaluate the efficacy of oral antifungal (voriconazole) as an adjuvant treatment to topical medication in severe filamentous fungal keratitis found no added benefit. However, a secondary sub-analysis from the MUTT-2 trial [68] showed a possible advantage of adding oral voriconazole to culture positive fusarium keratitis with reduced rate of perforation (but this was not statistically significant), a decreased the need for TPK, reduced scar size and better visual acuity at three months. Systemic antifungals, as an adjunct treatment to topical agents, are indicated in ulcers >5 mm in size, with involvement of >50% stromal depth, recalcitrant infections, bilateral infections, when associated with scleritis, with limbal involvement or endophthalmitis, paediatric cases, post keratoplasty infections and in cases of impending perforation/perforated ulcers with better healing experience in different case reports and in some case series [69,70,71]. Thus, given the inconclusive role of oral antifungals in fungal keratitis, more randomized control trials in this area will perhaps be helpful.

#### Challenges in Azole- and Polyene-Based Therapy

Despite their broad spectrum of activity, the physicochemical properties of azoles and polyenes, emergence of resistance, cross resistance, systemic and ocular toxicity and lack of appropriate AFST limit their potential to varying extents. Development of resistance to anti-fungal agents is an emerging challenge and the degree of susceptibility to resistance has been found to be greater with azoles than the polyenes. Resistance to polyenes is usually produced by increasing the synthesis of other sterols, leading to the emergence of fungal species with reduced ergosterol content in the cell membrane [72]. Other mechanisms reported include enhanced catalase activity, replacement, reorientation, and/or masking of some or all the polyene-binding sterols with those sterols with lower affinity to polyenes [73,74]. Resistance to the azole group of drugs occurs through multiple mechanisms and is more complex than that of the polyene group. The mechanisms are mainly overexpression of active efflux pumps, which results in a decreased concentration of the drug within the fungal cells; mutations in the ERG11 gene that encodes the lanosterol C14α-demethylase—the target enzyme for the azoles leading to poor binding of the agent to the enzyme; the upregulation of the ERG11 gene and also mechanisms of abnormal sterol synthesis in place of ergosterol, which, as seen with polyenes, can occur [73,75]. This resistant species of organism can lead to non-healing fungal ulcers and can lead to poorer visual and anatomical results.

Another major limitation is the lack of availability of commercial antifungal drugs and the need to reconstitute most of the agents (except for natamycin) from the parenteral formulations which are available. The ocular pharmacokinetics of these agents are another restriction, with most of the drugs failing to reach the desired concentration within the eye. The high molecular weight of these agents (amphotericin B, ketoconazole, miconazole) combined with poor water solubility (natamycin, itraconazole) contributes to poor intra-ocular penetration [76,77]. Voriconazole has higher bioavailability and is effective with systemic and local illness (topical, intrastromal) but has side effects like visual disturbances (hallucinations), colour vision disturbances and increased sensitivity to light [78]. Overcoming these challenges or finding effective newer agents to combat the resistant fungi with appropriate sensitivity-checking could help achieve better treatment outcomes in future.

### 3.3. Targeted Therapy

Targeted drug delivery in the form of intrastromal and intracameral injection of antifungal agents is a preferred adjunctive treatment in mycotic keratitis. This is due to presumed theoretical advantages, like delivering a steady-state concentration of drugs throughout the day, avoiding sub-optimal therapeutic levels and allowing better penetration especially in keratitis with deep stromal involvement. The commonly used agents are voriconazole at a dose of 50–100 µg/0.1 mL and amphotericin B at a dose of 5–7.5 µg/0.1 mL. Intrastromal, intracameral and intravitreal use of antifungal agents has been found useful in the treatment of recalcitrant keratitis [79,80], fungal keratitis with associated endophthalmitis [81] and also for post keratoplasty, kerato-refractive surgery and mycotic keratitis [82,83]. In the past, intrastromal and intracameral amphotericin B was preferred to treat recalcitrant fungal ulcers [80,84]. Owing to its higher incidence of ocular and systemic side effects (surface toxicity, renal toxicity), it is largely being replaced by the safer alternative of voriconazole, a second-generation azole with good activity (low MIC values) against common fungal pathogens like *Fusarium* spp. and *Aspergillus* spp. Multiple case series have showed favourable outcomes with the addition of intrastromal voriconazole to the standard treatment regimen [79,85,86]. However, Narayana et al.’s [87] randomized control study, assessing the efficacy of addition of intrastromal voriconazole 1% to a treatment regimen for moderate to severe filamentous fungal keratitis, found no added advantage in terms of day three and day seven culture positivity, final scar size, 3 month visual acuity or reduced rate of perforation/reduced need for TPK with the addition of intrastromal voriconazole when compared with topical natamycin 5% monotherapy. A recent study [88] comparing the safety and efficacy of the intrastromal injection of voriconazole, amphotericin and natamycin showed all three groups to be comparable in efficacy with the amphotericin group in comparison to the other two, which require more injections and also result in a large scar size with deep vascularization as a complication. The conflicting results between randomized control trials done by Narayana et al. [87] and Saluja et al. [88] have been attributed to variations in the fungal isolates obtained in each study with *Fusarium* predominating in the former and *Aspergillus* in the latter. Further randomized control trials in this direction need to be performed to verify the dose, dosing interval, minimum and maximum number of injections, indications, efficacy and safety of this mode of treatment.

## 4. Antifungal Agents

### 4.1. Natamycin

Introduced in the 1960s, this polyene macrolide has stood the test of time and is the most evidence-based medication currently available for the management of filamentous fungal keratitis. It is the primary drug of choice for most of the filamentous fungal infections of the eyes and acts by binding to sterols (primarily ergosterol) present in the fungal cell membrane to cause membrane instability leading to the death of the fungus. It is commercially available at a concentration of 5% in solution (50 mg/mL) and has a wide spectrum of activity involving mainly *Fusarium* and *Aspergillus* spp., and also *Alternaria*, *Candida*, *Cephalosporium*, *Colletotrichum*, *Curvularia*, *Lasiodiplodia*, *Scedosporium*, *Trichophyton* and *Penicillium* spp. [89]. One of the major limitations of natamycin has been its poor ocular penetration which prompted clinicians to seek drugs with better ocular penetration to achieve faster resolution of symptoms in keratitis.

This encouraged the use of voriconazole 1%, a drug belonging to the azole group which acts by inhibiting ergosterol synthesis which is an important constituent of the fungal cell membrane and theoretically has a better ocular penetration profile than natamycin. Despite the proposed superiority of voriconazole, natamycin therapy was found to have better outcomes in many of these studies. The largest of these, the MUTT trial [90], a double-masked multicentric randomized control trial (*n* = 323) performed comparing topical natamycin (5%) with voriconazole (1%), showed natamycin to be associated with better clinical and microbiological outcomes in smear-positive filamentous fungal keratitis especially in cases of fusarium keratitis. The rate of corneal perforation and the need for penetrating keratoplasty was higher in the voriconazole group when compared to the natamycin cohort [90]. Similar results were obtained in another randomised control trial [91] (*n* = 118) with the natamycin group having faster healing and better final visual acuity than the voriconazole group for filamentous fungal keratitis especially fusarium keratitis. A Cochrane review [92] of the efficacy of various antifungals also showed natamycin to have better clinical outcomes as compared to voriconazole.

Intrastromal use of natamycin is less explored and not commonly used due to its inherent characteristics like poor water solubility and ocular bioavailability. Experimental studies done in a rabbit model failed to show any benefit with the addition of intrastromal natamycin suspension to the treatment regimen [93]. The need for a better natamycin formulation with greater ocular bioavailability and reduced dosage led to the novel natamycin solution with better ocular bioavailability developed by Velpandian et al. [94] by combining natamycin with hydroxylpropylbetacyclodextrin (HPβCD). Natamycin solution at a concentration of 1% for topical use and 0.01% for intrastromal injection was evaluated for efficacy and toxicity in rabbit models and was found non-inferior to a 5% commercially available natamycin suspension. A clinical study [88] comparing the intrastromal use of this solution form of natamycin and other commonly used agents like voriconazole and amphotericin B showed this formulation to have significantly better mean healing time than the other two groups. Further studies with larger sample sizes need to be undertaken in this regard for better validation of this novel natamycin solution.

### 4.2. Amphotericin B

Amphotericin B is a broad-spectrum antifungal polyene macrolide produced by *Actinomyces Streptomyces nodusus* and was the first clinically used antifungal agent. Amphotericin B is fungicidal in action and by its interaction with ergosterol in the fungal cell membrane causes cell death by the formation of pores and by creating lethal membrane permeability changes. Its spectrum of activity mainly involves *Candida* spp., *Aspergillus* spp. and *Cryptococcus* and it is less effective against *Fusarium* spp. and other *Mucorales* with high MIC values [95]. The conventional route of administration is topical at a concentration of 0.15% (1.5 mg/mL) to 0.5% (5 mg/mL) solution but is limited by the need for access to a compounding pharmacy for the preparation of the desired concentration, poor ocular penetration and side effects like cytotoxicity at high concentrations leading to punctate corneal erosions, epithelial defects, stromal oedema and iritis [96,97]. Owing to its interaction with cholesterol in human cells, amphotericin B is associated with many side effects like nephrotoxicity, hepatotoxicity, erythema and hyperaemia of skin [96] when used systemically and hence is not a preferred systemic antifungal agent. The drug is available in two formulations, the conventional deoxycholate amphotericin B and newer lipid-based preparations like liposomal amphotericin B and lipid complex amphotericin B with a lesser toxicity profile and better ocular permeability. The experimental study [98] done comparing conventional amphotericin B with liposomal amphotericin B in rabbit models showed a faster and better response in the liposomal group with lesser features of ocular toxicity. However, these observed differences were not statistically significant and need further validation with experimental and human studies.

Many studies have recently emphasised the use of intracameral [80,84], intrastromal [80,99,100] and intravitreal amphotericin B [81] coupled with topical treatment alone or in combination with other standard drugs for better treatment outcomes. Conflicting results can be seen in literature regarding the intracameral use of amphotericin B. An earlier study in 2007 [84] reported it to be of benefit with faster response while a later randomized control trial [101] showed no added advantage of intracameral amphotericin B over topical antifungal therapy. Another study showed the combination of intracameral amphotericin B with intrastromal amphotericin B in severe recalcitrant fungal keratitis can give good treatment outcomes [80]. The intrastromal injection of amphotericin B has been found to be useful in indolent keratitis caused by *Candida* and other species. A higher rate of deep vascularization on healing was noted with intrastromal amphotericin B in a recent [88] comparison of intrastromal voriconazole, amphotericin B and natamycin. The relatively smaller number of patients and adjunctive use of other topical or systemic antifungal agents preclude a convincing conclusion on intrastromal usage.

Subconjunctival injection, the other route, which attempts to increase drug compliance and concentration, is not favoured due to the resulting conjunctival granuloma formation [102], necrosis and scleritis [103]. Successful use of subconjunctival amphotericin B without any complications in combination with topical amphotericin B and other standard anti-fungal agents has also been reported [104,105]. In an attempt to increase the ocular penetration of amphotericin B, a recent experimental study [106] evaluated the use of a liposomal amphotericin B microneedle ocular patch designed in a contact lens model done ex vivo and in vivo in rabbit models showing with promising results.

### 4.3. Voriconazole

Voriconazole is a newer generation azole which acts by inhibiting 14 a-lanosterol demethylase and affecting ergosterol synthesis, an essential component of fungal cell walls. It is US-FDA approved for invasive aspergillosis and its ophthalmic use, even though frequent, is off-label. It is available in both oral and parenteral formulation and the topical solution is made at a concentration of 1% (1 mg/mL) by the reconstitution of the parenteral drug. It is a broad-spectrum antifungal agent with activity against many fungi including *Aspergillus* spp., *Candida* spp., *Fusarium* spp., *Scedosporium* and *Cryptococcus* spp., but with minimal action against *Mucorales* [95,107].

Topical and intrastromal use are the two commonest modes of administration of this agent. Many studies have demonstrated topical formulation to have excellent activity against many common pathogenic fungi [108,109]. The largest randomized control trial, the MUTT-1 trail [90], comparing topical 1% voriconazole and natamycin 5%, failed to demonstrate its benefit over conventional, FDA-approved natamycin drops. However, with the available literature and from our personal experience, voriconazole seems to be the alternative drug for recalcitrant cases not responding to natamycin and amphotericin B and also as an adjunctive in severe fungal keratitis. Even though the randomized control trial [87] evaluating the role of intrastromal voriconazole application in fungal keratitis failed to demonstrate any benefit, voriconazole is the most commonly used intrastromal antifungal agent with many authors [79,85,88] reporting beneficial treatment outcomes with 50–100 µg/0.1 mL intrastromal injections.

Oral voriconazole is used in situations where keratitis is associated with endophthalmitis or scleritis and when convention therapy fails. The MUTT-2 trial [14] failed to show any added advantage of oral voriconazole in the treatment regimen but a secondary analysis [68] of this data showed it to have beneficial adjunctive role, but not statistically significant, in case of *Fusarium* keratitis. Sharma et al. [110] compared oral voriconazole with oral ketoconazole in keratitis cases and found better therapeutic efficacy with voriconazole. A recent case series [111] reported treating fungal keratitis with only voriconazole (200 mg BD) to be effective in eliminating the organisms and suggested it as a new potential weapon in the prophylaxis and management of fungal keratitis. Further randomized multicentric studies in this regard from different parts of the world are required to reach a conclusion.

### 4.4. Itraconazole

Itraconazole is a synthetic dioxolane triazole antifungal agent which has similar mechanism of action as other triazoles. The spectrum of activity includes *Candida* spp., *Aspergillus* spp. [112] and synergistic activity or additive interactions have been found when combined with natamycin 5% in *Fusarium* keratitis [113]. Itraconazole is available in both solution and ointment forms for ophthalmic use at a concentration of 1% (10 mg/mL) and at a systemic dose of 200–400 mg/day orally [114]. The systemic usage of this drug is limited due to side effects such as gastrointestinal upset, headache, transient skin reactions and, rarely, hepatitis [115]. Topical formulation also has limitations due to the lipophilic nature of the drug and its resultant poor ocular bioavailability. Multiple experimental studies done to overcome these limitations by fabricating itraconazole into nanosized carriers, like niosomes [116], microemulsion [117], nanosuspension [118], solid–lipid nanoparticles [119], nanovesicles-spanlastics [120] and nano-crystals (ITZ-NC) [121,122], have shown these formulations to have enhanced antifungal activity with better absorption than normal drug delivery systems. A recent study [122] incorporating itraconazole nano crystals into a thermosensitive in situ ocular gel base composed of a mixture of Pluronic^®^ F127, Pluronic^®^ F68 and hydroxylpropylmethylcellulose (HPMC) polymers found it to improve the pharmacokinetic profile of itraconazole over topical drops of ITZ-NC.

### 4.5. Posaconazole

Posaconazole is a second-generation triazole antifungal agent which has shown promising results against yeasts, many moulds and a few endemic fungal species [123]. It is US-FDA approved for used in oropharyngeal candidiasis and as prophylaxis for invasive *Aspergillus* and *Candida* infections in high risk individuals >13 years of age [123]. It is considered a key anti-fungal agent of the future due to its broad spectrum of activity and low MIC values [124]. Posaconazole is used at a dosage of 800 mg/day oral suspension (200 mg in four divided doses or 400 mg in two divided doses) systemically and hourly, or two-hourly at a concentration of 10 mg/0.1 mL or 4 mg/0.1 mL topically [125]. It is a well-tolerated drug with minor side effects like nausea, vomiting, diarrhoea, abdominal pain and rarely hepatocellular damage with elevated liver enzymes [126]. Multiple case reports [125,127,128,129,130] have documented its effectiveness in rare fungal infections and recalcitrant keratitis not responding to standard treatment regimens involving voriconazole, amphotericin, fluconazole and natamycin. The structural variation in the form of an extended sidechain in the posaconazole molecule as compared to voriconazole might help posaconazole in remaining effective even in cases resistant to voriconazole [131]. Altun et al. [129] reported two cases of fusarium keratitis which responded rapidly to oral (200 mg in four divided doses) and topical posaconazole (4 mg/0.1 mL) in a case unresponsive to oral and topical voriconazole and fluconazole. The spectrum of activity of posaconazole thus involves filamentous fungi like *Fusarium* spp. [125,129,130], *Aspergillus* spp., *Candida* spp. (even fluconazole-resistant ones), and rare fungi like *Paecilomyces* spp. [127] and *Beauveria* spp. [130]. A recent case series by Ferguson et al. [132] showed that high-dose oral posaconazole (500–600 mg once daily) provides rapid response to fungal keratitis (two cases of *Fusarium* and one case of *Paecilomyces*) not responding to conventional agents. No significant systemic side effects were found in this series with high-dose posaconazole therapy. Currently, posaconazole is used off label in many resistant and refractory ocular fungal infections affecting both the anterior and posterior segment. Experimental studies [133] in the field of nanoparticle drug delivery for posaconazole in the form of posaconazole micelles have shown these to have a better ocular permeation profile than the diluted oral suspension formulation but further in vivo studies are required to substantiate this evidence. Randomised control trials to optimize the concentration of topical posaconazole (10 mg/0.1 mL vs. 4 mg/0.1 mL) and to investigate its effectiveness as a monotherapy versus combination therapy and as a first-line agent need to be undertaken for better utilization of this drug.

### 4.6. Ketoconazole

This lipophilic imidazole acts by inhibiting 14a-sterol demethylase affecting ergosterol synthesis and is used as an oral tablet at a dose of 200–400 mg/day in two divided doses for fungal infections of the eye [134]. The spectrum of activity is rather narrow when compared to other azole drugs and is active against candida, other moulds and not much effective against the common pathogenic filamentous fungi [135]. The hepatotoxic side effect of this drug is the major limiting factor and hence regular monitoring of liver function tests is mandatory while on treatment [135]. Some reports suggest 60–80% healing of fungal ulcers with the use of oral ketoconazole [134,136]. Rajaraman et al. [137] conducted a randomized control trial to study the adjunctive role of oral ketoconazole with topical natamycin 5% and found no added benefit. Sharma et al. [110] also showed ketoconazole to be less effective than oral voriconazole. In view of efficacy reports and side effects associated with ketoconazole treatment, its systemic use in fungal keratitis is limited.

### 4.7. Luliconazole

The need for a better antifungal agent from the azole group with lower MIC for common pathogenic fungal organisms like the *Fusarium* species (voriconazole has a higher MIC against *Fusarium* spp.) has urged ophthalmologists worldwide to explore many new antifungal agents, used in other fungal infections of the body, in an ophthalmic setting. Luliconazole is a new imidazole antifungal agent with broad spectrum activity which is currently being used as 1% cream, 1% ointment and 1% solution in the topical treatment of dermatomycoses and onychomycoses [138]. An in vitro study [139] undertaken analysing the MIC of luliconazole against keratitis-derived *Fusarium* spp. and other reference strains showed luliconazole to have the lowest MIC value against all tested filamentous fungi and MIC_90_ of 0.06 μg/mL against *Fusarium* spp. Hence, further studies, both experimental and clinical for evaluation of this potential new antifungal agent as a treatment option in a topical formulation for filamentous fungal keratitis need to be conducted.

### 4.8. Echinocandins

This is a relatively new antifungal group of drugs that act by inhibiting the synthesis of (1,3)-D-glucan, one of the essential elements of fungal cell walls, causing apoptosis and the death of the fungi [140]. The selective action of the agent on fungal cell walls helps to reduce toxic side effects in the human body and makes it theoretically effective against strains that are resistant to azole and polyene antifungal agents. Caspofungin, micafungin and anidulafungin, the US-FDA-approved echinocandins, are currently used in the management of invasive systemic fungal infections, commonly caused by Candida and Aspergillus species [140]. The commercially available echinocandin formulation is for parenteral use with poor ocular absorption. The success of these drugs in invasive fungal diseases have urged ophthalmologists to explore its effect in ocular eye infections like keratitis and endophthalmitis. The spectrum of activity includes mainly *Candida* (fungicidal due to the high glucan content) [141] and *Aspergillus.* (fungistatic) and has been found to be less effective against *Fusarium* spp. [142]. Several in vitro and in vivo studies have shown promising results with respect to the safety, efficacy and dosage of these agents [143,144]. The studies done in an animal model [145] as well as in human volunteers [146] on the ocular penetration of caspofungin have shown that its penetration is better in an inflamed cornea or if the epithelial layer is denuded. Caspofungin used in concentrations of 1%/0.5% in solution for topical use and intravenous (IV) formulation for systemic use have been reported to have an adjunctive role in the management of ocular fungal infections caused by *Alternaria* [147,148], voriconazole refractory *Candida albicans* keratitis [149] and recurrent *Candida parapsilosis* keratitis [150]. Spriet et al. [151] have successfully used caspofungin (IV) along with voriconazole (IV) and posaconazole (IV) to treat endophthalmitis caused by *Aspergillus* and *Fusarium oxysporum*. Micafungin is the second drug in this category which has been researched for ophthalmologic use. Topical ophthalmologic solution 0.1% (1 mg/mL) has been found to be useful in recalcitrant yeast-related corneal ulcers [152], comparable to fluconazole 0.2% drops in *Candida* keratitis [153].

### 4.9. Other Therapeutic Agents

Some disinfectant agents being used as surface antiseptics in general ophthalmology practice with broad spectrum activity against bacteria, yeasts, moulds and certain viruses have recently been evaluated at lower concentrations for topical use as antimicrobial solutions. Pinna et al. conducted in vitro studies [154,155] to assess the activity of hexamidine diisethionate 0.05% (Keratosept) povidone-iodine 0.6% (IODIM^®^) and found it to have antimicrobial activity against *Candida* spp.

## 5. Nanoparticles

Nanotechnology involves dealing with particles in the size range of 1–100 nm and in the field of ophthalmology this has been explored in various aspects in relation to novel drug delivery systems (NDDS) and gene delivery. Nanoparticles in the form of nanosuspensions, liposomes, micelles, nanofibers, nanotubes, etc., are being utilised for the delivery of anti-fungal agents in an attempt to achieve better ocular penetration, retention and improved bioavailability. Several in vitro and in vivo experimental studies [118,119,121,156] have shown impressive results with these nano formulations but require further clinical studies for validation in humans.

## 6. Contact-Lens-Based Drug Delivery

A therapeutic contact lens is an ideal drug delivery system which can continuously provide a drug to an infected cornea while limiting non-specific absorption and the loss of the drug through tears and can play a significant role in the management of infectious keratitis. A contact lens drug delivery system for a fungal keratitis patient can be a big advantage as the arduous dosing regimen with lesser antifungal agents in the market often result in poor compliance to treatment. Some studies [157] in this field have been promising but currently there is no commercially available contact lens drug delivery system for fungal keratitis.

## 7. Photo Activated Chromophore for Keratitis

Corneal collagen crosslinking is a well-established procedure conventionally used in the management of ectatic corneal disorders like keratoconus and pellucid marginal degeneration and also in conditions like bullous keratopathy. The mechanism of action of photo-activated chromophore for keratitis (PACK-CXL) involves ultraviolet-A (UV-A) irradiation of the tissue primed with photosensitizer riboflavin (vitamin B2), resulting in the formation of reactive oxygen species and singlet oxygen which help to increase corneal biomechanical stability by forming covalent bonds between the stromal collagen fibrils. Schnitzler et al. in 2000 were the first to demonstrate the use of collagen crosslinking for erosive/ulcerative conditions of the cornea, with a remarkable outcome [158]. Recently, there has been an increase in the use of collagen crosslinking as an adjunct in the management of infective keratitis. The proposed mechanisms of action include (i) anti-inflammatory and anti-microbial action by causing damage to the DNA/RNA of the pathogens, (ii) increased resistance to enzymatic degradation of the tissue by organisms [159] and (iii) enhancement of the ocular penetration of antifungals with the help of corneal collagen crosslinking [160]. Some experimental studies [159,160,161] have shown corneal collagen crosslinking to be an effective adjunctive treatment to the existing algorithm for managing fungal keratitis, with increased advantage when combined with anti-fungal agents at an early phase in the disease course. A prolonged duration of irradiation [162] and a higher concentration of riboflavin [160] were also proposed to increase the efficacy of the procedure. However, the role of corneal crosslinking in the management of mycotic keratitis remains controversial in humans with different studies providing conflicting results. Some case reports [162,163,164,165] and major studies [159,166,167] have shown corneal crosslinking to be a useful adjunctive therapy for managing fungal keratitis. However, the Cross Linking Assisted Infection Reduction Trial (CLAIR trial), a randomised control trial [159] done to evaluate the role of corneal collagen crosslinking in the management of fungal keratitis concluded that it has no added advantage and furthermore suggested that crosslinking can result in inferior visual outcomes when compared with standard medical management in fungal keratitis.

## 8. Photodynamic Therapy

Photodynamic therapy (PDT) involves the formation of reactive oxygen species (ROS) from photosensitizers when activated with a source of radiation and these ROS damage cell structure and result in cell death. Conventionally, PDT is used in the treatment of choroidal neovascularization, polypoidal choroidal vasculopathy, tumours of the choroid and rarely for lens epithelial cell proliferation and corneal neovascularization. PDT has been proposed as an alternative treatment modality for corneal infections like acanthamoeba and fungal keratitis [168]. PDT appears to be a promising treatment due to its obvious advantages like negligible drug resistance, high spatiotemporal control, and lesser side effects when compared to available topical medications. The protocol followed is similar to that of riboflavin with UV-A used in keratoconus cases. Experimental studies [169] done comparing rose Bengal versus riboflavin PDT showed rose Bengal PDT with green light (518 nm) to be significantly effective in vitro against common species of fungal pathogens like *Fusarium solani*, *Aspergillus fumigatus*, *Candida albicans*. The close evolutionary relationship between the mammalian cells and the fungus has always been a matter of concern with many side effects occurring due to similarity in structure. An innovative recent study [170] utilised mitochondrial-targeting luminogens with aggregation-induced emission characteristics (AIE gens) as photosensitizers in place of rose Bengal and riboflavin, which will selectively target the fungal mitochondria, the organelle which has the least potential to develop drug resistance. This comparative study [170] done in a rabbit model compared these AIE gens, namely IQTPE-2O, IQ-Cm and IQ-TPA-assisted PDT with rose Bengal PDT and found it to be less toxic and more efficacious.

## 9. Surgical Management

Surgical treatment is an important modality in treating fungal keratitis with 50% [14] of cases requiring therapeutic keratoplasty for the control of infection (Figure 6). In a study done in eastern India [67], it was seen that 51.58% of filamentous fungal keratitis cases (43.33% *Aspergillus* and 60.86% *Fusarium*) needed TPK for the control of infection with 100% of *Candida* cases requiring TPK. The need for TPK in mycotic keratitis ranges from 15% to 55%, suggesting that medical treatment alone may not be successful in the management of fungal keratitis [171,172,173]. The common indications for TPK in a case of fungal keratitis include perforated ulcers, impending perforations and recalcitrant keratitis worsening on conservative management [171]. Prajna et al. [13], in a secondary analysis with the MUTT-2 data, evaluated the predictors that could potentially indicate the need for a TPK in fungal keratitis. The presence of hypopyon along with infiltrate size and depth were noted to have a significant association with the need for TPK. The presence of hypopyon indicated 2.28 times the odds and a 1 mm increase in infiltrate size on follow-up indicated 1.74 times the odds in favour of corneal perforation/the need for TPK. Infiltrates involving the posterior 1/3rd of the cornea were associated with a 71.4% risk for perforation/need for TPK [13]. The survival of a corneal graft in an actively infected and inflamed eye is very minimal with an increased chance of graft rejection, re-infection and secondary glaucoma [171]. The rates of re-infection after keratoplasty procedures are a major concern in fungal keratitis with rates ranging from 6% to 16% [171,173,174]. The scarcity of donor corneas in developing countries in comparison to the demand is a huge challenge in the management of mycotic keratitis cases requiring TPK. Another important reason for rapid graft failure and poor outcomes is the lack of availability of optical-grade tissues with good endothelial counts in developing countries where non-optical grade tissue is used for TPK given the increased demand and paucity of donor corneal tissue. Further, poorer outcomes from surgical intervention in fungal keratitis cases can occur due to the delayed use of steroid drops post keratoplasty (fear of re-infection) and hence an increase in inflammation and resultant graft decompensation and vascularisation [171].

### Lamellar Keratoplasty in Fungal Keratitis

Lamellar keratoplasty (LK) in managing fungal keratitis is gaining popularity due to the obvious advantages of a lamellar procedure over a full-thickness procedure and is being performed in many places in selective cases. Xie et al. [175] in their study comprising 55 fungal keratitis cases refractory to conventional treatments, LK gave a success rate of 92.7% (*n* = 51) with a recurrence rate of just 7.3% (*n* = 4) and good visual and anatomical outcomes after 12–18 months of follow-up. Various other studies [26,176] have also concluded early surgical intervention in the form of deep anterior lamellar keratoplasty (DALK) can be a safe therapeutic approach in the management of fungal keratitis. A recent study [177] evaluated the efficacy of DALK in fungal keratitis with crosslinked acellular porcine corneal stroma followed by topical tacrolimus therapy and reported good surgical and visual outcomes at the end of one year of follow up.

## 10. Conclusions

Fungal keratitis is common in warm, humid regions with varying geographic profiles. The virulence and bioburden of the pathogenic fungi, host defence mechanism and immune response and difficulty in diagnosis and treatment play a significant role in the resulting outcome. In the setting of conventional diagnostic approaches (clinical diagnoses, smears and cultures) failing to provide reliable diagnosis, refractory mycotic keratitis can result in poor prognosis. In vivo confocal microscopy for fungal hyphae detection is helpful to confirm diagnosis when smear and culture detection does not yield results, with molecular diagnostics methods gaining popularity. Antifungal susceptibility testing to establish the sensitivity to treatment with conventional and newer antifungal agents can help in cases of refractory fungal keratitis. Recently recommended omics approaches, such as those using genomic, metagenomic and tear proteomic data sources, seem to provide greater hope for better diagnosis and follow-up of fungal keratitis cases. A recent metagenomic approach is based on 16S rRNA genes to help monitor the dynamic change in conjunctival microbial flora associated with a fungal keratitis episode. Diagnostics based on 18S rRNA target enrichment sequencing can serve to diagnose fungal corneal infections using clinical samples. MDS allows for rapid diagnosis and is more effective for obtaining an accurate diagnosis without the need to wait for the fungus to grow. The custom tear proteomic approach is also evolving as an important modality in the diagnostics and management of mycotic keratitis. Rapid detection by multiplex PCR and antifungal susceptibility testing is being suggested as a routine management protocol of fungal keratitis to improve treatment outcomes. The positivity of repeat cultures at six days following antifungal therapy helps to identify cases at a higher risk for therapeutic keratoplasty, worse three-month visual acuity and larger scar size. Repeat cultures are seen as a useful tool for prognosticating and identifying fungal keratitis that might benefit from a therapeutic keratoplasty performed early and are now considered helpful for establishing the efficacy of new antifungal agents in comparison to time to healing or visual outcome. Newer antifungal agents and combination therapy in comparison to monotherapy are seen to be more efficacious in the management of mycotic keratitis. Future strategies to mitigate morbidity due to mycotic keratitis will need to be multidimensional, with adjuvant therapies aimed at modifying the immune response to infection to improve clinical outcomes.

## Figures and Tables

**Figure 1 jof-07-00907-f001:**
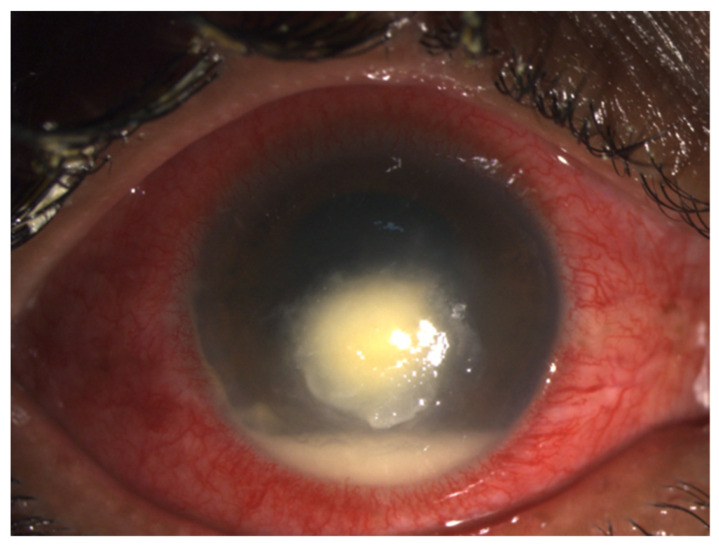
Clinical photograph of a 30 year old female patient with history of trauma to right eye showing central dry-looking corneal ulcer with feathery margins and hypopyon of 1.2 mm in size suggestive of fungal keratitis.

**Figure 2 jof-07-00907-f002:**
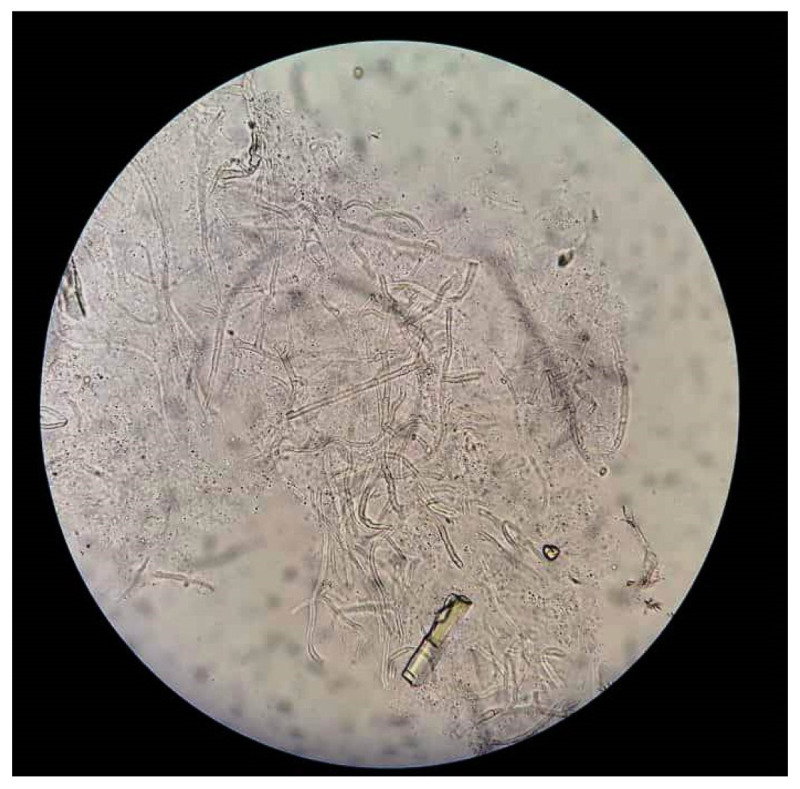
KOH mount of corneal scraping in a case of fungal keratitis showing the branching hyphae.

**Figure 3 jof-07-00907-f003:**
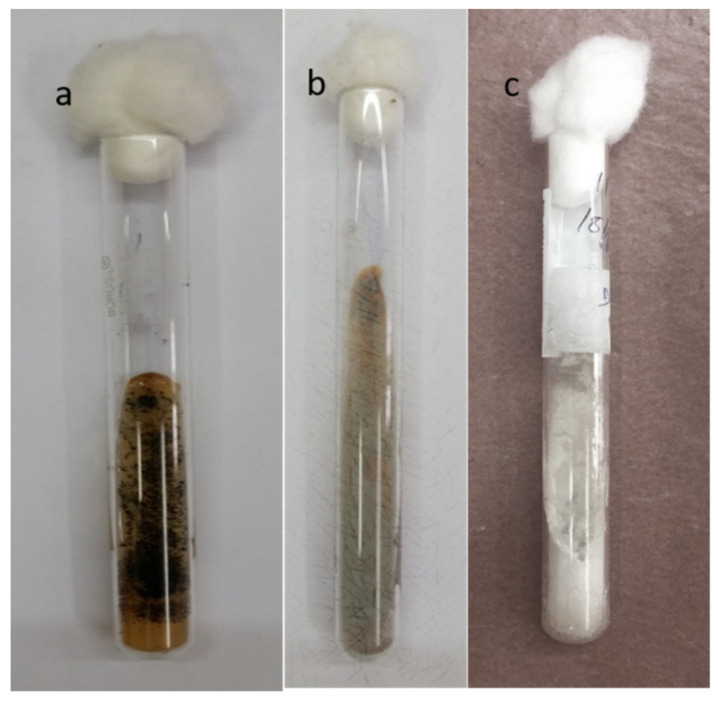
Sabouraud’s Dextrose Agar showing growth of moulds of various colours (**a**) black colonies of *Aspergillus niger.* (**b**) slate green colonies of *Penicillium* spp. (**c**) cream colonies of *Fusarium* spp.

**Figure 4 jof-07-00907-f004:**
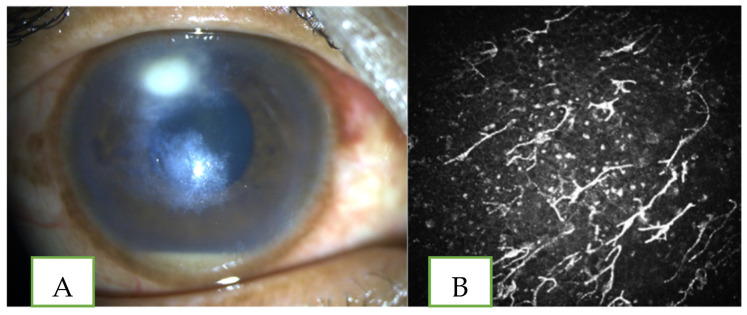
Clinical photograph (**A**) in a case of non-healing keratitis with 2 satellite areas of deep infiltrates and 0.5 mm hypopyon (**B**) IVCM pictures of the patient showing branching refractile elements suggestive of fungal hyphae.

**Figure 5 jof-07-00907-f005:**
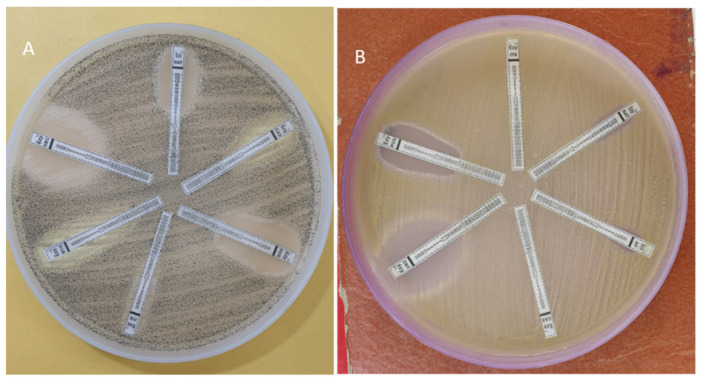
Anti-fungal susceptibility testing using E test in (**A**) *Aspergillus niger* (**B**) *Aspergillus flavus*.

**Figure 6 jof-07-00907-f006:**
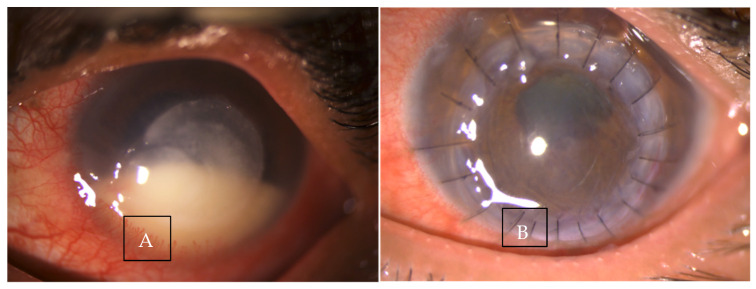
Clinical photograph (**A**) in a case of recalcitrant fungal keratitis with impending perforation (**B**) showing post operative day 1 picture post therapeutic keratoplasty.

## Data Availability

Not applicable.

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
