# Peer review of "Recent Perspectives in the Management of Fungal Keratitis"

_jof, 2021, doi:10.3390/jof7110907_

Round 1

Reviewer 1 Report

The review is in good preparation, easy to follow and extensive. The authors made the review of conventional techniques for diagnosis of fungal keratitis, and the currently available molecular methods to aid rapid diagnosis and help in epidemiological study. The provided information on treatment is very useful for management of fungal keratitis.

The manuscript should be improved as followings.

  1. Relevant references are not included to provide sufficient support. For example, P2, L58-59 (in-intro confocal microscopy and molecular diagnosis …..and real-time diagnosis). P3,L96-98 abd P7, L161-162, reference for confocal microscopy should be added. P3, L101-104, references for example of some techniques used for aiding direct examination such as calcofluor white and some tissue staining should be included. P7, L235-237 (recent sophisticated metagenomics approach…with fungal keratitis)
  2. Figure 2 should be replaced with the better focusing plane for hyphal visualization.
  3. Writing mistakes are seen scattered throughout the entire manuscript, especially full stop that should be placed after reference bracket. The other mistakes included,
  • Unnecessary hyphen (For example, P 1; L10 and 37; P14, L620).
  • Spacing, especially between number and concentration unit and between word and reference bracket, at entire the manuscript.
  • Unnecessary italics, e.g. P2,L51 and other places.
  • Fungal names were forgotten to be italicized in some place, e.g. Fig3 and 5 legends; P5, L179,180; P7,L270; P8,L293; P12, L512; P13, L548; P14, L599,614; P16,L704).
  • In vivo, in vitro, ex vivo, in situ should be italicized.
  • Inconsistent format for fungal name. Some of Genera followed with species or spp but some are not. Spp should have full stop at after.
  • Full name for abbreviation are missing, e.g. PK, P60,L61; DALK, P4, L152; BD, P12, L477; CXL, P15, L664; RCT, P15, L673; LK should be added on P17.
  • Mistake in bolding (ref. 4,13,22,23,26,27 in P5)
  • The word “(molds)” (P7,L268) should be either moved to place after filamentous fungi or deleted.
  • P8,L293; Amphotericin B (the A should not be capitalized).
  • Font difference (P8,L312-315).
  • P9,L323, should be 0.15% amphotericin for yeast-like fungi.
  • P9,L482-483. Mistake at “actinomyces” and “Streptomyces nodusus”
  • P14, L629, L is missing from mL
  • P15, L651, CXL should be removed. Line652 "(" close bracket sign has to be removed.
  1. Antifungal agents should be reordered, the polyenes then azoles.

Author Response

We thank the reviewer for taking out valuable time to enhance our manuscrip twith his/her kind comments.

the point wise reply is enclosed in the uploaded file

Reviewer 2 Report

Dears authors 

Analysis by paper partitions:   1 - Introduction: it must be reformed in the content and in the writing of the general part to review the syntax of the topic   2- Discussion: deepen (max two lines) the use of the new disinfectants that prevent mycotic keratitis, since there are few therapeutic weapons available. Learn more about this by using and citing the following references: PMID: 32452982; PMID: 31486592 ; PMID: 31626916   3 - Check the bibliographic entries throughout the text, some of which are non-compliant, review some entries in the bibliographic references and necessarily insert those referred to in point 2 for the purpose of acceptance by me.   4 - Review the English grammar and in particular the applied scientific English: in particular, the verb tenses and the syntax in the discussion.

Author Response

we thank the reviewer for his kind comments. our point wise reply is enclosed in the file uploaded

Round 2

Reviewer 2 Report

The corrections on been made.